# Influence of Strain Rate on Plastic Deformation of the Flange in Steel Road Barrier

**DOI:** 10.3390/ma16041396

**Published:** 2023-02-07

**Authors:** Martin Pitoňák, Ján Ondruš, Katarína Zgútová, Miroslav Neslušan, Ján Moravec

**Affiliations:** 1Faculty of Civil Engineering, University of Žilina, Univerzitná 1, 01026 Žilina, Slovakia; 2Faculty of Operation and Economics of Transport and Communications, University of Žilina, Univerzitná 1, 01026 Žilina, Slovakia; 3Faculty of Mechanical Engineering, University of Žilina, Univerzitná 1, 01026 Žilina, Slovakia

**Keywords:** plastic deformation, strain rate, Zn layer, heating

## Abstract

This study investigates the influence of strain rate on plastic deformation developed in the flange of a steel road barrier. This effect can be investigated by the use of the uniaxial tensile test. It was found that the strain rate increases yield as well as ultimate strength and gently drops down the elongation at break. Moreover, the accelerated strain rate is connected with matrix heating and increasing the Taylor–Quinney coefficient. Despite the valuable matrix heating and the higher Taylor–Quinney coefficient at the higher strain rates, samples necking is initiated earlier and dislocation density is higher. Flange grains become preferentially aligned along the direction of uniaxial stress, especially at the higher plastic strains. Finally, surface Zn protective layer delamination is initiated quite early beyond the yielding. It is considered that the cracks are due to the different response of the Zn allayer and underlying steel matrix on the plastic straining. Increasing strain rate attenuates the degree of Zn layer delamination.

## 1. Introduction

Steel road barriers are very important with respect to the safety of road traffic. These barriers bear random impacts during traffic accidents in which a variety of angles, car masses and/or velocities are considered [1,2,3]. The main technical requirements for the different degrees of safety are clearly introduced in the national standards [1]. These national standards follow the international standard EN 1317 which defines common testing and certification procedures for Road Restraint Systems [4]. The EN 1317-5 standard serves as the framework for the marking of road safety systems such as barriers, crash cushions, barrier terminal ends and transitions [5].

The reliable operation of these barriers depends on many aspects such as their installation (especially grounding), the type of flange, their maintenance, the degree of corrosion damage, etc. [6,7,8,9]. For instance, a corrosion attack can remarkably decrease the effective cross-section of a body which, in turn, decreases its bearing capacity and redistributes the stress to the neighbouring regions [10]. For this reason, all steel components of the barrier are subjected to the Zn galvanizing process to avoid premature failure [11,12]. Barrier grounding is of vital importance especially in hilly roads, on bridges and/or high-speed roads [1,2,3].

The initial impact of the vehicle is borne by the flange and its plastic deformation stores the majority of kinetic energy *E_k_*. This energy can be calculated as follows:(1)$$E_{k}=0.5m\;{\left(v\;sin\varphi \right)}^{2}$$
where *m* is a mass of vehicle, *v* is the vehicle speed, and *φ* is the angle of collision [1]. The bearing capacity of a flange can be simulated on the basis of its shape, wall thickness, and the aforementioned collision parameters (*m*, *v* and *φ*), as well as mechanical properties. Mechanical properties of the matrix can be extracted from the stress–strain curves obtained from the standardized uniaxial tensile tests [13]. However, two basic aspects should be reported as critical.

Stress–strain curves usually refer to the uniaxial tensile load, whereas the true plastic behaviour of a flange can be biaxial or multiaxial.Tensile tests are usually carried out at the standardized strain rate which can be far away from the real one.

Oversimplifications in materials testing can result in failures of systems in their real operation. Conditions of pure tensile stresses on the flange during real impacts can be found when the plastic deformation of the impacted flange is redistributed to the neighbouring one. The high deformation speed and the corresponding high strain rate due to the high velocity of a vehicle *v* can be simulated during tensile testing with the varied strain rate. Such testing can be routinely carried out on modern testing systems which can perform tensile and other tests. Equipment of these devices can be customized in respect of specific requirements. Tests can be performed within a certain range of strain rates and plastic strains, at the elevated temperatures, etc.

It is known that a certain volume of plastic work is converted into heat beyond the yielding which, in turn, results in matrix heating. This behaviour can be analysed by the use of the Taylor–Quinney coefficient *(β_INT_*) [12,13]. This coefficient can be derived from the first law of thermodynamics taking into consideration the adiabatic conditions as follows [14,15]:(2)$$\beta_{INT}=\frac{\rho C_{p}\Delta T}{\int dW_{p}}$$
where *W_p_* is incremental plastic work, *ρ* refers to the density, *C_p_* heat capacity of the deformed matrix, and Δ*T* is the temperature growth. *β_INT_* expresses how much plastic energy is converted to heating and which fraction is directly stored in the matrix in the form of lattice imperfections [14,15]. It is well known that the Taylor–Quinney coefficient *β_INT_* is usually less than 1 in the cases when other heat sources are not involved [16]. Low values of *β_INT_* indicate increasing density of lattice imperfections, whereas *β_INT_* near 1 indicates a quite stable structure when all the plastic work is converted to heat. Rittel et al. [16] reported that *β_INT_* is higher for Ti alloys as compared with Al or Fe alloys. Zaera et al. [17] found increased *β_INT_* during plastic straining of austenitic steels due to phase transformation. Smith [18] reported thermal softening during plastic deformation at high strain rates employing the Taylor–Quinney coefficient. Soares et al. [14] as well as Rittel [15] investigated the influence of strain rate on *β_INT_* in the high entropy alloys and the glassy polymers.

This study reports the influence of strain rate on the plastic behaviour of the steel matrix employed in the flanges of steel road barriers employing the aforementioned Taylor–Quinney coefficient *β_INT_*. Moreover, related aspects of plastic deformation such as microhardness evolution, residual stress state, mechanical properties, and Zn practice layer delamination are discussed as well. The response of flanges with respect to their mechanical load is investigated in laboratory conditions via the uniaxial tension test. Flanges in real operation are mostly loaded by bending. However, their load is redistributed to the neighbouring flanges in which the uniaxial tension dominates. The main goal of this research is focused on the monitoring of flanges which are not visibly altered in shape but their bearing capacity is altered due to transmission of the energy during collision from the bended flanges. These flanges could be potentially used for further operation when their bearing capacity is not decreased valuably.

## 2. Experimental Conditions

Samples for the uniaxial tensile test were cut from the new flange NH4 of wall thickness 3 mm. The shape and dimensions of samples for the uniaxial tensile test are shown in Figure 1. The final shape and the corresponding size (see Figure 1) were obtained by the milling process with application of a coolant in order to avoid the thermal overload of the samples. The uniaxial tensile tests were carried out in accordance with the ISO standard 6892-1. However, the strain rate as well as the length of the necked region were altered in order to model the true load of flanges when in real operation. The flange was Zn-galvanized and the average thickness of the Zn coating was 135 μm. Further details associated with the Zn layer (its heterogeneity), the chemical composition of the Zn layer, as well as the underlying matrix, together with the image of the microstructure, can be found in the previous study [19]. This study also provides information about the flange shape and the region from which the samples for the tensile test were cut off. The microstructure of the flange is fully composed of ferritic equiaxed grains and a limited volume of pearlite island (metallographic images provided in the next text). The direction along the flange length is referred to as the rolling direction (RD), whereas the transversal direction along the flange width is abbreviated TD, see Figure 1. The direction along the flange wall thickness is referred to as ND.

An Instron 5985 device (Instron, Norwood, USA) was used for uniaxial plastic straining, and the true elastic strains were measured by the 2620-602 extensometer (Instron, Norwood, USA). Apart from the investigation of full stress–strain curves until breakage, a series of the samples were loaded until reaching the predefined plastic strains as indicated in Table 1. Four strain rates were employed, 0.5 × 10^−3^ s^−1^, 2.0 × 10^−3^ s^−1^, 8.0 × 10^−3^ s^−1^, and 32.0 × 10^−3^ s^−1^. The study investigates sensitivity of plastic behaviour against the strain rate. For this reason, the strain rate was altered in the wide range. The low strain rates are closer to the conventional tensile test with respect to ISO 6892-1, whereas the highest one is closer to the real collisions. The highest strain rate was set on the base of the preliminary recommendations of the Instron device manufacturer in order to avoid its improper usage and possible damage. Three repetitive samples were carried out for each strain and strain rate (60 samples). Instron data were sampled at the frequency of 100 Hz.

*W_p_*, incremental plastic work for the analysis of *β_INT_*, was calculated on the basis of the loading force during the tensile test (data list provided by the Instron system) and the true strain (taking into consideration the initial gauged length 40 mm). The values *ρ* = 7850 kg.m^−3^ and *C_p_* = 452 J.kg^−1^ K^−1^ were used for calculations of *β_INT_* with respect to Equation (1). Temperature *T* was measured on the sample in situ of the tensile test using industrial temperature sensor PPG101A6 (Mouser electronics, Munich, Germany) in software Tera Term (Tera Term Project, Tokyo, Japan), (sampling frequency also 100 Hz). The temperature sensor was fixed in the centre of the necked region (with respect to its length as well as width, see Figure 1) by the tape. The surface of the sensor as well as the samples were greased by the heat conductive paste in order to maintain good heat transfer between the sample and the sensor.

In order to investigate matrix alterations in the different plastic strain, specimens of length 20 mm were cut along RD and routinely prepared for metallographic observations (hot moulded, ground, polished and etched by 3% Nital for 5 s). Apart from metallographic observations which provide the cross-sectional view in the RD-ND perspective, also RD-TD views were obtained in order to study the degree of Zn layer delamination within the gauged length. RD-TD views were obtained using the light microscopes Zeiss AxioCam MRc5 (Zeiss, Oberkochen, Germany) and Olympus SZx16 in Quick Photo Industrial 3.0 software (Olympus, Tokyo, Japan).

Finally, the X-ray diffraction (XRD) technique was employed for the measurement of residual stresses and evaluation full width at half maximum (*FWHM*) of the diffraction peak which is usually linked with strain-hardening degree and the corresponding dislocation density, in this particular case [20]. Residual stresses and *FWHM* in the predefined strains (as indicated in Table 1) were measured after unloading in the RD as well as the TD by the Proto iXRD Combo diffractometer (Proto Manufacturing Ltd., Montreal, Canada) (*Kα1* and *Kα2* of {211} planes, CrKα, Winholtz and Cohen method, ½s_2_ = 5.75 TPa^−1^, s_1_ = −1.25 TPa^−1^).

## 3. Results of Experiments and Their Discussion

### 3.1. Mechanical Properties

The influence of higher strain rates is analysed since the real vehicle impacts are very often at higher speeds and the corresponding high strain rates. For this reason, the mechanical properties of the flange matrix can be different from the nominal ones or those measured at lower strain rates [17,21,22]. Figure 2 illustrates the engineering stress–strain curves for the different strain rates. These curves exhibit typical evolution for the low alloyed steels of *bbc* lattice and the limited volume of soluble elements when upper and lower yield points together with Luder’s strain region can be recognized [13,23]. Dislocation motion and their intersection can be considered as the prevailing mechanism of strain-hardening. As soon as the upper yield point is attained, dislocation motion is initiated and plastic deformation becomes developed. Initially, the dislocation slip in the Luder’s region runs under the lower yield point. However, the Luder’s region is soon replaced by increasing stress *σ_y_* along with increasing strain when the mutual interaction of dislocations can be considered. The evolution of strain hardening can be expressed by the well-known Hall–Petch equation:*σ_y_*= *σ_i_* + *k_y_ d*^−1/2^(3)
where *σ_i_* is the friction stress of the lattice, 2*d* is the average grain diameter (*d* is the average grain radius), and *k_y_* is the parameter depending on the grain size.
*k_y_*= *m*^2^ *τ_c_ r*^1/2^(4)
where *τ_c_* is shear stress needed to unpin a dislocation, *r* is the distance from the pile-up to the neighbour source, and *m* is the factor which is a function of the relative direction of shear stress *τ* against normal stress *σ*. Equations (3) and (4) clearly indicate that the higher value of the lower yield point at a higher strain rate (as that shown in Figure 2 and indicated in Table 2) is mainly due to the higher *σ_i_* as a result of the higher velocity of dislocation motion (see Equation (5)) because the average grain diameter as well as *k_y_* are nearly the same early beyond the yielding for all strain rates. Strain rate *γ* is a function of dislocation velocity *ν* as follows:γ = *ρ*_*m*_
*b**ν*(5)
where *ρ_m_* is the density of dislocations and *b* is the Burgers vector [13,23]. Expressed in other words, acceleration of dislocation motion at higher strain rates increases the pinning strength of local stress fields around the lattice imperfections.

Table 2 and Figure 2 also demonstrate the following:-the length of the Luders region is shorter at lower strain,-yield point as well as ultimate strength grow along with the increasing strain rate,-and, finally, elongation at break drops down, along with increasing strain rate.

### 3.2. β_INT_ Analysis

Strain rate has no valuable influence on the evolution of *W_p_* along strain, see Figure 3. *W_p_* grows nearly linearly with strain. However, the valuable difference in sample heating can be found, as illustrated in Figure 4 and Figure 5. Δ*T* at strain rate 0.5 × 10^−3^ s^−1^ is only 2.1 °C, at strain rate 2.0 × 10^−3^ s^−1^ it is about 7.6 °C, at strain rate 8.0 × 10^−3^ s^−1^ it’s 20.4 °C, and finally at strain rate 32.0 × 10^−3^ s^−1^ it’s 24.3 °C. Figure 4 also illustrates that the initial increase of *T* becomes delayed along with the increasing strain rate. Increasing temperature indicates that the increasing fraction of *W_p_* is consumed on sample heating which, in turn, means that the accelerated velocity of dislocations results in accelerated oscillations of encountered atoms.

The evolution of *β_INT_* for the different strain rates is remarkably different with respect to values as well as the evolution along with the strain, see Figure 6. Bearing in mind nearly the same *W_p_* for all strain rates, the difference in *β_INT_* should be linked to differences in sample heating, as illustrated in Figure 4. Thus, the higher sample heating strongly correlates with higher *β_INT_* and vice versa, see also Figure 7. Apart from *β_INT_* values, also the evolution of the Taylor–Quinney coefficient with strain is different for the higher strain rates as contrasted against the lower ones. *β_INT_* exhibits continuous growth for strain rates 8.0 × 10^−3^ s^−1^ as well as 32.0 × 10^−3^ s^−1^, whereas the evolution of *β_INT_* for strain rate 2.0 × 10^−3^ s^−1^ is more or less flat. Finally, the evolution of *β_INT_* for strain rate 0.5 × 10^−3^ s^−1^ exhibits progressive descent. An explanation of these differences should be found in heat transfer from the gauged region of the width 14 mm towards the neighbouring region of the width 20 mm, see Figure 1. Calculation of the Taylor–Quinney coefficient considers an adiabatic process, in which the heat produced in the gauged region is not transferred into the neighbouring ungauged regions. However, such conditions can be met at higher strain rates only when the duration of the tensile test and the corresponding heating is short-time. As soon as the plastic strain is prolonged at lower strain rates, the validity of the model applied for the calculation of the Taylor–Quinney coefficient is disturbed [14].

### 3.3. XRD Measurements

Figure 8 depicts the release of the tensile residual stresses in RD as well as TD. The initial residual stresses before the tensile test are developed by cold rolling of the flange and the consecutive Zn galvanizing at elevated temperatures. Bulk stresses are around 100 MPa in both directions. Plastic straining produces compressive stresses in RD of low amplitude compensated by the low amplitude tensile residual stresses in TD. Differences among the strain rates are limited and become more valuable for plastic strains of 32.5% only when the amplitude of compressive stresses increases along with the strain rate in RD as well as TD. Moreover, gently higher *FWHM* in RD and TD can be found at higher strain rates at plastic strain 32.5%, see Figure 9. This figure also demonstrates that this evolution is reversed at plastic strain 5%. As aforementioned, the *FWHM* of the XRD peak can be directly linked with matrix hardness and the corresponding dislocation density [20,24,25]. One might expect that *FWHM* for higher strain rates at the end of the test would be lower due to the thermal softening effect with respect to the increasing sample heating and the higher *β_INT_* [26]. However, the measured temperatures are too low and play only a minor role with respect to dislocation density [13]. The main reason can be found in the more developed true strains linked with a steeper decrease of the engineering stress–strain curves at the higher strain rates, see Figure 2 (especially the difference for plastic strains 32.5%). Plastic strain 32.5% is quite early before the sample breakage which means more developed matrix straining in the necked region for the higher strain rates [27]. On the other hand, sample breakage for the lower strain rates is delayed which, in turn, means less developed strain-hardening. This behaviour can also explain the evolution of residual stresses in RD and TD for the plastic strain 32.5%.

### 3.4. Metallographic Observation and Zn Layer Delamination

The metallographic images, as illustrated in Figure 10 and Figure 11, demonstrate that the visible grain elongation cannot be found at lower plastic strains and becomes valuable only before breakage. More developed plastic straining depicts Figure 10d and Figure 11d. These figures clearly depict that the degree of preferential grain elongation is more developed at the higher strain rate. The degree of grain elongation is more developed for the higher strain rates due to more developed true strains as well as the synergistic contribution of higher temperatures [28].

On the other hand, Figure 10, Figure 11 and Figure 12 illustrate the delamination of the Zn layer from the sample surface. The Zn layer remains on the surface but contains longitudinal and cross-sectional cracks, see Figure 10a and Figure 12b. These cracks are related to bimodal and triaxial strain development in the materials. It is considered that the cracks are due to the different response of the Zn allayer and underlying steel matrix on the plastic straining. Further sample straining results in Zn layer delamination. However, the remains of the Zn layer still can be found on the sample surface since Zn layer delamination is developed along the longitudinal cracks, see Figure 10a,b and Figure 12c. Presence of the Zn remains of the surface indicates the tough binding of Zn layer with the underlying body through the diffusion layer [11,19]. Further plastic straining reduces the fraction of these remains (see Figure 10c and Figure 12d). Finally, remains of the free surface can be found at plastic strains 26.5 and 32.5%, see Figure 10d and Figure 12e,f. However, the near surface region still contains the valuable content of Zn in the diffusion layer [19,29].

Figure 11 demonstrates that the fraction of the Zn layer that remains is greater at higher strain rates as compared with the lower one depicted in Figure 10. It is considered that heating the sample makes the Zn coating more malleable, which delays its delamination. For this reason, the sample strained to ε = 32.5% at strain rate 32.0 × 10^−3^ s^−1^ still contains a thin Zn layer, see Figure 11d.

On one hand, Zn layer delamination might be an unwanted phenomenon with respect to corrosion resistance of a flange in use. On the other hand, it was reported that the criterion for the acceptable bearing capacity of flanges exerted to plastic deformation more or less overlaps with initiation of Zn layer delamination [30]. For this reason, Zn layer delamination can be considered as the alternative criterion for rejection of flanges from the point of view of their bearing capacity.

## 4. Conclusions

The increasing Taylor–Quinney coefficient usually indicates that the higher fraction of energy consumed during the plastic work is converted into the heat at the expense of lower density of lattice imperfections. However, in this particular case, such behaviour was not found and accumulation of dislocation tangles at the higher strain rates was enhanced at the same plastic strain. It should be also mentioned that the measured temperatures are too low for, and have nearly no influence on, dislocations (their annihilation or/and motion). This behaviour also confirms increasing yield and ultimate strength as well as the reduced elongation at break. Having nearly the same evolution of the consumed energy at the different strain rates, it can be considered that the dynamics of the neighbouring dislocations as well as the grains interaction plays the major role. Lower dislocation velocity at the lower strain rates delays the samples necking due to improved exchange interaction among the grains in the matrix.

## Figures and Tables

**Figure 1 materials-16-01396-f001:**
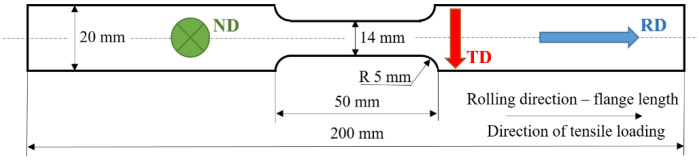
Shape and dimensions of the sample employed for the uniaxial tensile test.

**Figure 2 materials-16-01396-f002:**
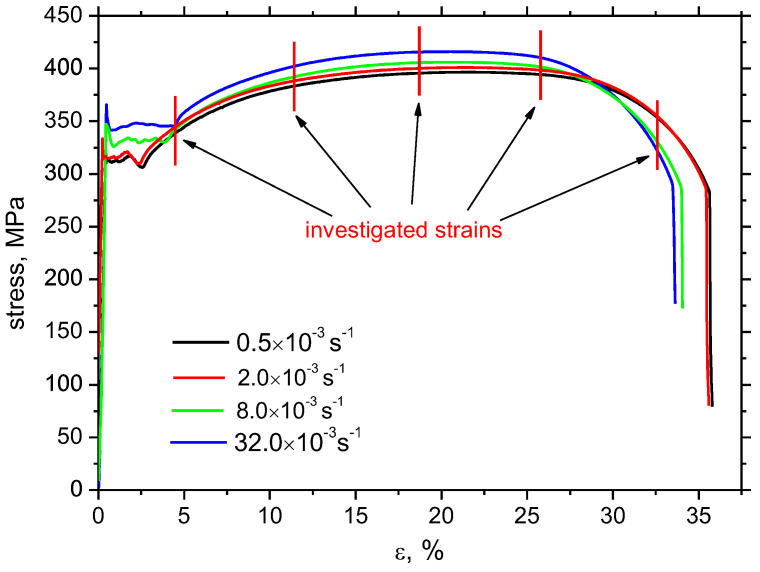
Stress–strain curves for variable strain rates.

**Figure 3 materials-16-01396-f003:**
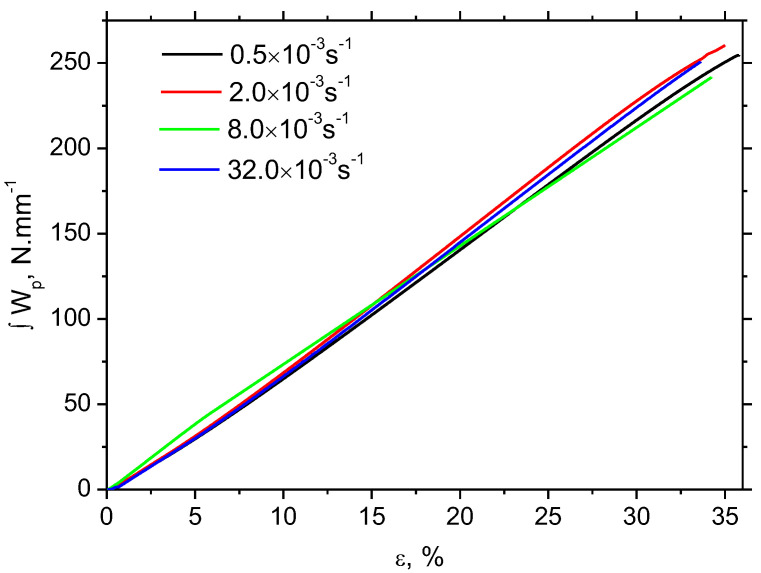
Evolution of plastic work *W_p_* as a function of plastic strain and strain rate.

**Figure 4 materials-16-01396-f004:**
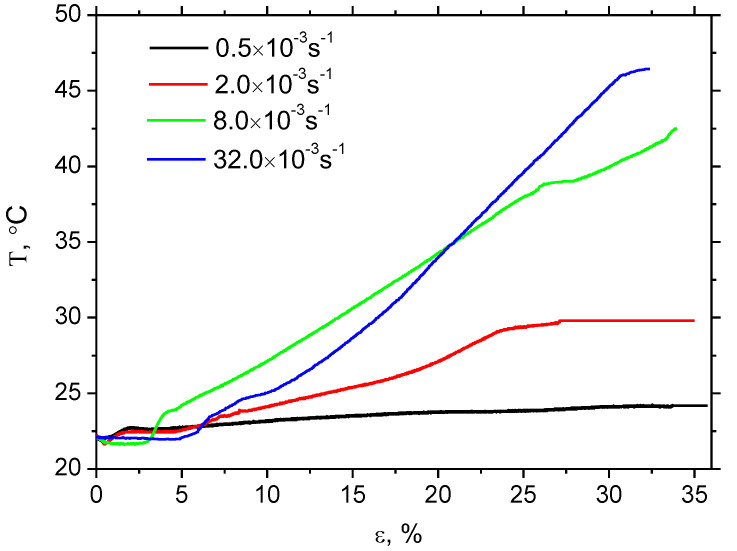
Evolution of temperature *T* as a function of plastic strain and strain rate.

**Figure 5 materials-16-01396-f005:**
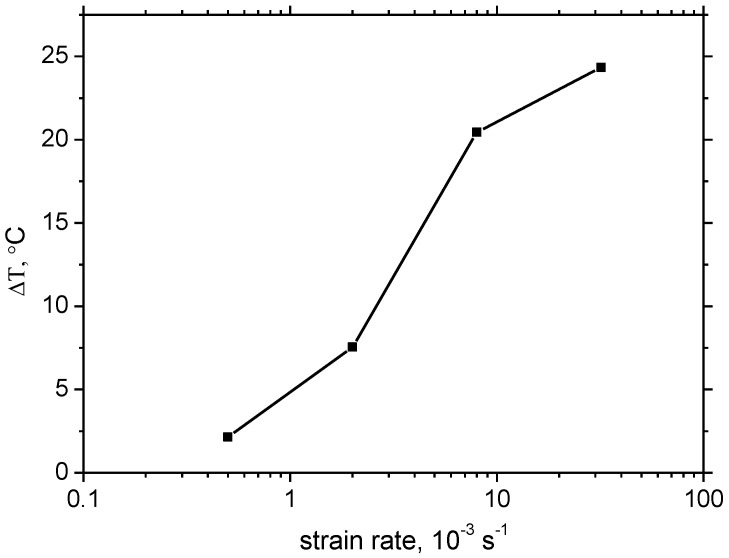
Δ*T* as a function of strain rate at plastic strain 32.5%.

**Figure 6 materials-16-01396-f006:**
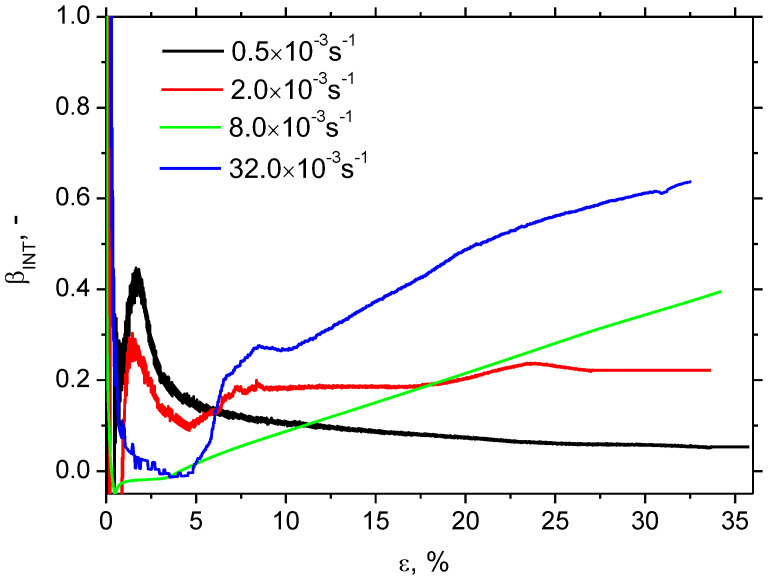
Evolution of *β_INT_* as a function of plastic strain and strain rate.

**Figure 7 materials-16-01396-f007:**
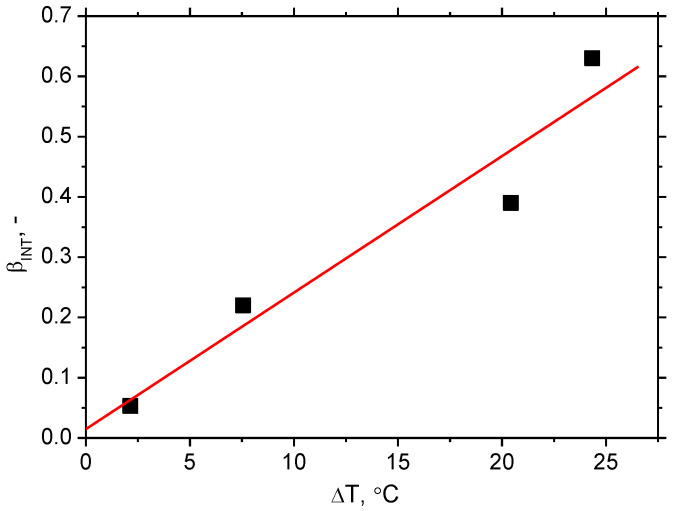
Evolution of *β_INT_* as a function of plastic strain and strain rate at the end of the test.

**Figure 8 materials-16-01396-f008:**
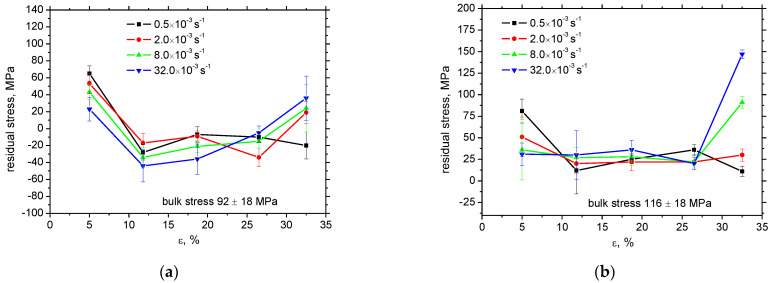
Evolution of residual stresses as a function of plastic strain and strain rate. (**a**) RD, (**b**) TD.

**Figure 9 materials-16-01396-f009:**
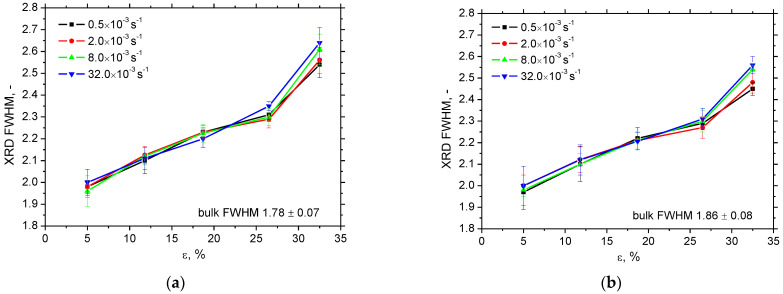
Evolution of XRD *FWHM* as a function of plastic strain and strain rate. (**a**) RD, (**b**) TD.

**Figure 10 materials-16-01396-f010:**
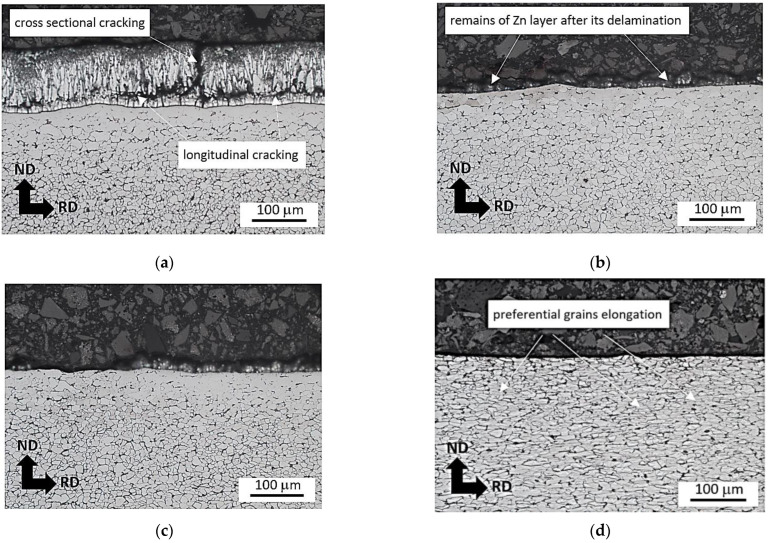
Metallographic images of the matrix and delamination of Zn layer during tensile test as a function of plastic strain for strain rate 0.5 × 10^−3^ s^−1^. (**a**) ε = 5.0%, (**b**) ε = 11.8%, (**c**) ε = 18.7%, (**d**) ε = 32.5%.

**Figure 11 materials-16-01396-f011:**
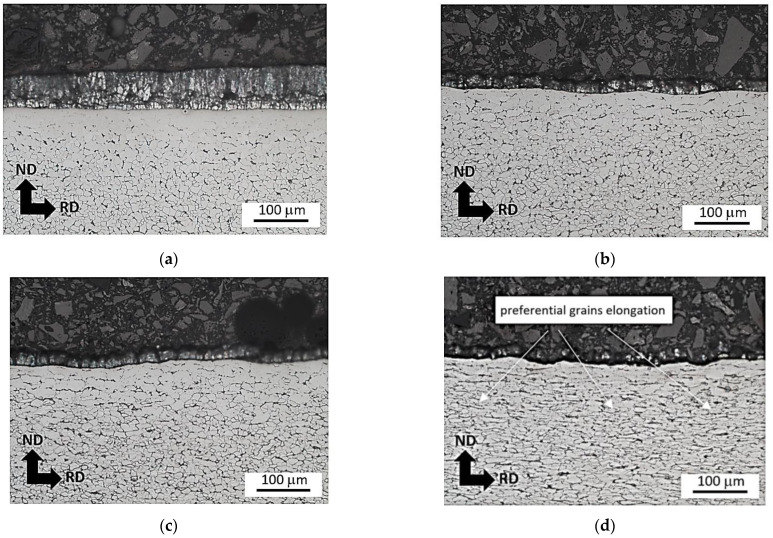
Metallographic images of the matrix and delamination of the Zn layer during tensile test as a function of plastic strain for strain rate 32.0 × 10^−3^ s^−1^. (**a**) ε = 5.0%, (**b**) ε = 11.8%, (**c**) ε = 18.7%, (**d**) ε = 32.5%.

**Figure 12 materials-16-01396-f012:**
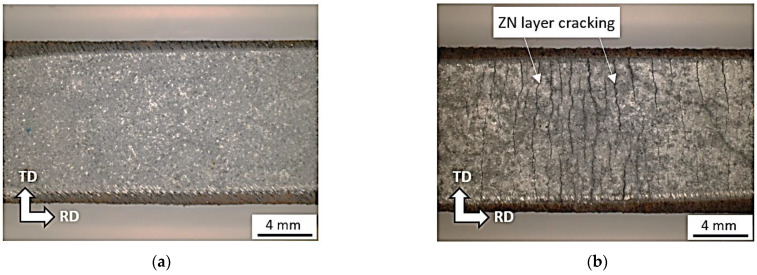
Delamination of the Zn layer during tensile test as a function of plastic strain for strain rate 0.5 × 10^−3^ s^−1^, RD-TD views. (**a**) bulk, (**b**) ε = 5.0%, (**c**) ε = 11.8%, (**d**) ε = 18.7%, (**e**) ε = 36.5%, (**f**) ε = 32.5%.

**Table 1 materials-16-01396-t001:** Predefined plastic strains.

	Homogenous			Non-Homogenous (Necking)	
strain	5%	11.8%	18.7%	25.6%	32.5%

**Table 2 materials-16-01396-t002:** Mechanical properties a function of strain rate obtained from 3 repetitive measurements.

Strain Rate, s^−1^	0.5 × 10^−3^	2.0 × 10^−3^	8.0 × 10^−3^	32.0 × 10^−3^
Lower yield point, MPa	317 ± 3	319 ± 3	329 ± 3	342 ± 3
Upper yield point, MPa	320 ± 3	333 ± 3	344 ± 3	362 ± 3
Ultimate strength, MPa	390 ± 3	404 ± 3	405 ± 3	415 ± 3
Elongation at break, %	35.1 ± 1.4	34.8 ± 1.5	33.2 ± 1.7	33.0 ± 1.3

## Data Availability

The raw data required to reproduce these findings cannot be shared easily due to technical limitations (some files are too large). However, authors can share the data on any individual request (please contact the corresponding author by the use of their mailing address).

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
