# Peer review of "Influence of Strain Rate on Plastic Deformation of the Flange in Steel Road Barrier"

_materials, 2023, doi:10.3390/ma16041396_

Round 1

Reviewer 1 Report

Comments and specific questions requiring clarification:

1.   I recommend not to divide the words. If it is possible please correct it. It applies to the whole manuscript.

2. Please use the full names when a given symbol or shortcut appears for the first time in the article (shortcut in brackets), and then in the rest of the paper the abbreviations can be used - please correct it in whole paper also in the abstract (bINT, FWHM, XRD).

3. Reffers to the line no. 13 and 14: There is information that “plastic deformation is also a function of strain rate” Explain it for me – what do You mean? I can agree that the plastic deformation (plastic strain) depends on strain rate – it is normal. But I don’t understand that the plastic deformation is also a function of strain rate?

4. Reffers to the line no. 27 and 28: “The main technical requirements for the different degrees of safety are clearly introduced in the national standards [1].” What about international standards. Please add information about it with literature references.

5. Reffers to the line no. 42: “collision parameters” - what parameters? -  please list them in detail and maybe characterize them.

6. Reffers to the line no. 54: “Such testing can be routinely carried out on modern testing systems”. What kind of modern testing systems do You mean – please add it if it is possible.

7. Whether the dimensions of the used samples comply with the standard for static tensile test  - add standard number. Moreover add unit of radius – 5 mm.

8. Reffers to the line no. 96: On what basis such strain rates were chosen? Add this information please. In my opinion the strain rates during traffic accidents is higher then You chose. Please provide justification.

9. Table 1 content is not chemical composition of the flange steel matrix. Correct it please. And explain what is this?

10. How was the temperature measured during the experiments?

Author Response

Reviewer: I recommend not to divide the words. If it is possible please correct it. It applies to the whole manuscript.

Response: We have checked ends of all lines and moved the divided words to the next line in the whole text.

Manuscript: Please check it in the manuscript.

Reviewer: Please use the full names when a given symbol or shortcut appears for the first time in the article (shortcut in brackets), and then in the rest of the paper the abbreviations can be used - please correct it in whole paper also in the abstract (bINT, FWHM, XRD).

Response: we edited the require corrections in the whole manuscript.

Manuscript: please check the manuscript.

Reviewer: Reffers to the line no. 13 and 14: There is information that “plastic deformation is also a function of strain rate” Explain it for me – what do You mean? I can agree that the plastic deformation (plastic strain) depends on strain rate – it is normal. But I don’t understand that the plastic deformation is also a function of strain rate?

Response: Of course, plastic deformation is function of strain rate and affect mutual interaction of dislocations and their free motion. Dislocation slip in a certain grain also trigger the similar process in the neighbouring grains and thermal aspects.

Manuscript: We added references associated with this aspect.

For this reason, the mechanical properties of the flange matrix can be different from the nominal ones or those measured at lower strain rates [17, 21, 22].

Reviewer: Reffers to the line no. 27 and 28: “The main technical requirements for the different degrees of safety are clearly introduced in the national standards [1].” What about international standards. Please add information about it with literature references.

Response: There is European standard EN1317-5. This standard defines common testing and certification procedures for Road Restraint Systems. Each country follow their internal technical documents developed on the base of the reasonable factors as well as with respect of EN1317-5. The national standards are therefore very often similar. The national technical documents are developed in accordance of EN1317-5.   

Manuscript:

These national standards follow the international standard EN 1317 which defines common testing and certification procedures for Road Restraint Systems [4]. The EN 1317-5 standard serves as the framework for the marking of road safety systems such as barriers, crash cushions, barrier terminal ends and transitions [5].

Reviewer: Reffers to the line no. 42: “collision parameters” - what parameters? -  please list them in detail and maybe characterize them.

Response: These parameters are indicated in the equation (1).

Manuscript: Being clear we altered the following sentence.

The bearing capacity of a flange can be simulated on the basis of its shape, wall thickness, and the aforementioned collision parameters (m, v and φ), as well as mechanical properties.

Reviewer: Reffers to the line no. 54: “Such testing can be routinely carried out on modern testing systems”. What kind of modern testing systems do You mean – please add it if it is possible.

Response: These systems are produced by many companies such as Instron, Zwick Roel, etc. Their equipment can be customized and the tests can be carried out at the different conditions depending on the purpose of the test.

Manuscript:

Such testing can be routinely carried out on modern testing systems which can perform tensile and other tests. Equipment of these devices can be customized with respect of specific requirements. Tests can be performed within a certain range of strain rates and plastic strains, at the elevated temperatures, etc. 

Reviewer: Whether the dimensions of the used samples comply with the standard for static tensile test  - add standard number. Moreover add unit of radius – 5 mm.

Response: We added the required information. We also added unit as required.

Manuscript: please check the appearance of Figure 1.

The uniaxial tensile tests were carried in accordance with the ISO standard 6892-1. However, the strain rate as well as the length of the necked region were altered in order to model the true load of flanges in the real operation.

Reviewer: Reffers to the line no. 96: On what basis such strain rates were chosen? Add this information please. In my opinion the strain rates during traffic accidents is higher then You chose. Please provide justification.

Response: We tried to investigate plastic deformation at low strain rates being close to the conventional tensile test and strain rate with respect ISO 6892-1 as well as the highest one being closer to the real collisions. The highest strain rate was set on the base of previous recommendations of the tensile device manufacture (in order to avoid its improper usage and possible damage). On the other hand, we agree these strain rates are lower as that expected in the real collisions. It is also worth to mentioned that the main reason of the study is analysis of the parts being loaded by uniaxial tensile stress. These flanges are connected with the flanges being loaded directly by bending during the collisions. The load is redistributed to them and the strain rate in these regions should be lower and closer to our strain rates.  

Manuscript:

The study investigates sensitivity of plastic behaviour against the strain rate. For this reason, the strain rate was altered in the wide range. The low strain rates are closer to the conventional tensile test with respect the ISO 6892-1 whereas the highest one is closer to the real collisions. The highest strain rate was set on the base of the preliminary recommendations of the Instron device manufacturer in order to avoid its improper usage and possible damage.

Reviewer: Table 1 content is not chemical composition of the flange steel matrix. Correct it please. And explain what is this?

Response: Yes, we agree. It is mistake. This table indicates the predefined plastic strains.

Manuscript:

Table 1. Predefined plastic strains.

Reviewer: How was the temperature measured during the experiments?

Response: We added further information.

Manuscript:

The temperature sensor was fixed in the centre of the necked region (with respect of its length as well as width, see Figure 1) by the tape. Surface of the sensor as well as the samples were greased by the heat conductive paste in order to maintain good heat transfer between the sample and the sensor.

Reviewer 2 Report

I like to thank the authors for crafting the manuscript. This paper reviews the influence of strain rate on plastic deformation developed in the flange of steel road barriers using uniaxial tensile testing only. Detailed comments are given below. I request the authors to address the following comments for the improvement of the manuscript:

1.      Line 13: Steel road barriers also experience bending stress during deformation. Why do authors not consider bending and combined stress and strain rate in the proposed investigation for this specific application in the flange of steel barrier?

2.      Line 17: What is βINT here? It should be defined when used first time in the manuscript.

3.      Line 17-19: so many information in a single sentence. The authors should consider splitting the sentence and providing single information in a sentence.

4.      Line 21: Add brief information about the Zn layer.

5.      Overall, the abstract is confusing and needs careful revision to summarize the work in brief with enough information to understand easily.

6.      Line 44-51: How is a simplification to uniaxial tensile done? Need better clarification in the explanation of this concept.

7.      Line 100: How do the authors calculate incremental plastic work?

8.      Line 221: Figures 10d and Figure 11d should be explained later than Figures 10 and 11 (a-c). A proper sequence in Figure should be followed.

9.      Line 221-223: Please indicate the preferential grain orientation in the figures.

10.  Line 226-228: What is the reason for longitudinal and cross-sectional cracks during longitudinal tensile strain? Is it related to bimodal and triaxial strain development in the materials?

11.  Although there are statements regarding dislocation density and residual stress, no result or the scientific discussion has been presented. Authors should link results with available literature.

12.  Conclusion: the section seems to be a brief repetition of results without discussion. The authors should provide scientific conclusions instead of a brief overview of the results.

13.  The quality of figures with descriptions needs substantial improvement.

14.  Justification for uniaxial tensile test in the context of application needs to be defined properly, and that should be addressed in the introduction section.

Author Response

All changes made in the manuscript (additional texts and corrections) are highlighted yellow colour (valid for the manuscript as well as this document).

Reviewer: Line 13: Steel road barriers also experience bending stress during deformation. Why do authors not consider bending and combined stress and strain rate in the proposed investigation for this specific application in the flange of steel barrier?

Response: We carried also 3 points bending load with application of strain-gauges. However, with respect of the main goal of the study and the real monitoring of flanges we decided to apply uniaxial tensile test. The manner in which the flange being neighbouring with the flange loaded by bending is usually uniaxial tension. The main goal of our studies is focused on the monitoring of plastic deformation and the possible further application of flange barriers for their further use. In reality check the flanges being deformed directly after collisions, loaded by bending cannot be used again due to their altered shape and size as well as bearing capacity. However, the load form these flanges is transmitted into the neighbouring flanges and these flanges are mostly loaded by uniaxial tension. The main task is monitoring of plastic deformation is these flanges with respect of their further use.   

Manuscript: We prefer no change.

Reviewer: Line 17: What is βINT here? It should be defined when used first time in the manuscript.

Response: We replaced this symbol by its full title Taylor-Quinney coefficient and we explain this parameter later.

Manuscript: Please check the abstract and Introduction part.

Reviewer: Line 17-19: so many information in a single sentence. The authors should consider splitting the sentence and providing single information in a sentence.

Response: we altered text – splitting the sentence as required.

Manuscript:

The mechanism of plastic deformation is based on dislocations motion and their mutual interaction. More developed plastic strain increases matrix hardness and associated Full Width at Half maximum of X-ray diffraction peak.

Reviewer: Line 21: Add brief information about the Zn layer.

Response: please check our previous study

Pitoňák, M.; Ondruš, J.; Minárik, P.; Kubjatko, T.; Neslušan, M. Magnetic Measurement of Zn Layer Heterogeneity on the Flange of the Steel Road Barrier. Mater. 2022, 15, 1898; doi.org/10.3390/ma15051898.

All important aspects were explained in this study. We refer this study and can be found in the list of references.

Manuscript: We prefer no change.

“Further details associated with the Zn layer (its heterogeneity), the chemical composition of the Zn layer as well as the underlying matrix together with the image of the microstructure can be found in the previous study [17]. This study also provides information about the flange shape and the region from which the samples for the tensile test were cut off.”

Reviewer: Overall, the abstract is confusing and needs careful revision to summarize the work in brief with enough information to understand easily.

Response: We altered the abstract.

Manuscript:

This study investigates the influence of strain rate on plastic deformation developed in the flange of steel road barrier. This effect can be investigated by the use of the uniaxial tensile test. It was found that the strain rate increases yield as well as ultimate strength and gently drops down the elongation at break. Moreover, the accelerated strain rate is connected with matrix heating and increasing the Taylor – Quinney coefficient. Despite the valuable matrix heating and the higher Taylor – Quinney coefficient at the higher strain rates, samples necking is initiated earlier and dislocation density is higher. Flange grains become preferentially aligned along the direction of uniaxial stress, especially at the higher plastic strains. Finally, surface Zn protective layer delamination is initiated quite early beyond the yielding.  It is considered that the cracks are due to the different response of Zn allayer and underlying steel matrix on the plastic straining. Increasing strain rate attenuates the degree of Zn layer delamination.

Reviewer: Line 44-51: How is a simplification to uniaxial tensile done? Need better clarification in the explanation of this concept.

Response: The concept in which the samples are loaded by tensile load is introduced in the last paragraph of section “introduction”. We also added some further details with respect of tensile test.    

Manuscript:

Response of flanges with respect of their mechanical load is investigated in the laboratory conditions via uniaxial tension test. Flanges in the real operation are mostly loaded by bending. However, their load is redistributed to the neighbouring flanges in which the uniaxial tension dominates. The main goal of this research is focused on the monitoring of flanges which are not visibly altered in shape but their bearing capacity is altered due to transmission of the energy during collision from the bended flanges. These flanges could be potentially used for further operation when their bearing capacity is not decreased valuably.

The final shape and the corresponding size (see Figure 1) were obtained by the milling process with application of a coolant in order to avoid the thermal overload of the samples. The uniaxial tensile tests were carried in accordance with the ISO standard 6892-1. However, the strain rate as well as the length of the necked region were altered in order to model the true load of flanges in the real operation.

Reviewer: Line 100: How do the authors calculate incremental plastic work?

Response: Calculation of incremental plastic work was based on the known load (force during the tensile test) and its multiplication by the corresponding sample elongation. These parameters were obtained from the Instron data report.

Manuscript:

This information is already presented in the original version of the manuscript. We prefer no change.

Wp, incremental plastic work for the analysis of bINT, was calculated on the basis of the loading force during the tensile test (data list provided by the Instron system) and the true strain (taking into consideration the initial gauged length 40 mm).

Reviewer: Line 221: Figures 10d and Figure 11d should be explained later than Figures 10 and 11 (a-c). A proper sequence in Figure should be followed.

Response: We altered the text.

Manuscript:

The metallographic images as illustrated in Figure 10 and Figure 11 demonstrate that the visible grain elongation cannot be found at lower plastic strains and becomes valuable only before breakage. More developed plastic straining depicts Figure 10d and Figure 11d.

Reviewer: Line 221-223: Please indicate the preferential grain orientation in the figures.

Response: We added indication of the preferential grains elongation for plastic strains 32.5% in Fig. 10d and 11d.

Manuscript: Please check appearance of Fig. 10d and 11d.

Reviewer: Line 226-228: What is the reason for longitudinal and cross-sectional cracks during longitudinal tensile strain? Is it related to bimodal and triaxial strain development in the materials?

Response: Might be this comments is true. It is quite difficult to verify this information. We consider that these cracks are developed due to the different response of Zn allayer and underlying steel matrix on the plastic straining.

Manuscript: We include the aforementioned comments in the corresponding text.

These cracks are related to bimodal and triaxial strain development in the materials. It is considered that the cracks are due to the different response of Zn allayer and underlying steel matrix on the plastic straining.

Reviewer: Although there are statements regarding dislocation density and residual stress, no result or the scientific discussion has been presented. Authors should link results with available literature.

Response: We added the related references.

Manuscript: Please check the related text.

Reviewer: Conclusion: the section seems to be a brief repetition of results without discussion. The authors should provide scientific conclusions instead of a brief overview of the results.

Response: We altered this section.

Manuscript:

The increasing Taylor – Quinney coefficient usually indicates that the higher fraction of energy consumed during the plastic work is converted into the heat at the expense of lower density of lattice imperfections. However, in this particular case such behaviour was not found and accumulation of dislocation tangles at the higher strain rates was enhanced at the same plastic strain. It should be also mentioned that the measured temperatures are too low for and have nearly no influence on dislocations (their annihilation or/and motion). This behaviour also confirms increasing yield and ultimate strength as well as the reduced elongation at break. Having nearly the same evolution of the consumed energy at the different strain rates, it can be considered that the dynamics of the neighbouring dislocations as well as the grains interaction plays the major role. Lower dislocation velocity at the lower strain rates delays the samples necking due to improved exchange interaction among the grains in the matrix.  

Reviewer: The quality of figures with descriptions needs substantial improvement.

Response: We improved quality of some figures. However, some figures we are not capable to improved substantially. We did our best. Might be the quality of our figures is altered during the conversion process from doc to pdf format.

Manuscript: Please check appearance of the figures through the manuscript as well as their descriptions.

Reviewer: Justification for uniaxial tensile test in the context of application needs to be defined properly, and that should be addressed in the introduction section.

Response: We added the required explanation of the concept in which uniaxial tension is applied in the study in the introduction section.

Manuscript:

Response of flanges with respect of their mechanical load is investigated in the laboratory conditions via uniaxial tension test. Flanges in the real operation are mostly loaded by bending. However, their load is redistributed to the neighbouring flanges in which the uniaxial tension dominates. The main goal of this research is focused on the monitoring of flanges which are not visibly altered in shape but their bearing capacity is altered due to transmission of the energy during collision from the bended flanges. These flanges could be potentially used for further operation when their bearing capacity is not decreased valuably.

Reviewer 3 Report

1.The literature review of this paper needs to be enrich, the number of references is too small. There is no sufficient summary of the existing research.

2.What is d in the formula (2)?

3.Is there any standard for the selection of test size?

4.How to select the range of strain rate, why increase by 4 times? If there is relevant literature reference, it needs to be cited correctly.

5.The formula (3) shows d-1/2 but line 136 shows 2d is the average grain diameter. Please verify.

6. The symbols of lines 5 and Line 146 are inconsistent.

7. Please check the ordinate of Figure 6.

8.The discussion in Section 3.4 can be further analyzed.

Author Response

All changes made in the manuscript (additional texts and corrections) are highlighted yellow colour (valid for the manuscript as well as this document).

Reviewer: The literature review of this paper needs to be enrich, the number of references is too small. There is no sufficient summary of the existing research.

Response: we added next references

Manuscript: please check it through the manuscript as well as list of references

Reviewer: What is d in the formula (2)?

Response: It is indicated in the text below, 2d is the average grain size. Therefore, d is the average grain radius.

Manuscript: we altered the corresponding sentence.

…the friction stress of the lattice, 2d is the average grain diameter (d is the average grain radius), and …

Reviewer: Is there any standard for the selection of test size?

Reviewer: How to select the range of strain rate, why increase by 4 times? If there is relevant literature reference, it needs to be cited correctly.

Response: We tried to investigate plastic deformation at low strain rates being close to the conventional tensile test and strain rate with respect ISO 6892-1 as well as the highest one being closer to the real collisions. The highest strain rate was set on the base of previous recommendations of the tensile device manufacture (in order to avoid its improper usage and possible damage). On the other hand, we agree these strain rates are lower as that expected in the real collisions. It is also worth to mentioned that the main reason of the study is analysis of the parts being loaded by uniaxial tensile stress. These flanges are connected with the flanges being loaded directly by bending during the collisions. The load is redistributed to them and the strain rate in these regions should be lower and closer to our strain rates. 

Manuscript:

The uniaxial tensile tests were carried in accordance with the ISO standard 6892-1. However, the strain rate as well as the length of the necked region were altered in order to model the true load of flanges in the real operation.

The study investigates sensitivity of plastic behaviour against the strain rate. For this reason, the strain rate was altered in the wide range. The low strain rates are closer to the conventional tensile test with respect the ISO 6892-1 whereas the highest one is closer to the real collisions. The highest strain rate was set on the base of the preliminary recommendations of the Instron device manufacturer in order to avoid its improper usage and possible damage.

Reviewer: The formula (3) shows d-1/2 but line 136 shows 2d is the average grain diameter. Please verify.

Response: it is correct.

Manuscript: We prefer no change.

Reviewer: The symbols of lines 5 and Line 146 are inconsistent.

Response: corrected

Manuscript: corrected

Reviewer: Please check the ordinate of Figure 6.

Response: Ordinates are correct.

Manuscript: We prefer no change.

Reviewer: The discussion in Section 3.4 can be further analyzed.

Response: we added next text as well as the associated references.

Manuscript:

The degree of grain elongation is more developed for the higher strain rates due to more developed true strains as well as the synergistic contribution of higher temperatures [28]. 

Presence of the Zn remains of the surface indicates the tough binding of Zn layer with the underlying body through the diffusion layer [11, 19]. Further plastic straining reduces the fraction of these remains (see Figure 10c and Figure 12d). Finally, remains free surface can be found at plastic strains 26.5 and 32.5%, see Figure 10d, Figure 12e,f. However, the near surface region still contains the valuable content of Zn in the diffusion layer [19, 29].

On one hand, Zn layer delamination might be unwanted phenomenon with respect of corrosion resistance of a flange in use. On the other hand, it was reported that the criterion for the acceptable bearing capacity of flanges exerted to plastic deformation more or less overlap with initiation of Zn layer delamination [30]. For this reason, Zn layer delamination can be considered as the alternative criterion for rejection of flanges from the point of view of their bearing capacity.  

Round 2

Reviewer 1 Report

Thank You very much for the answers. I accept the answers

Reviewer 2 Report

I thank the authors for modifying the manuscript as per the comments.

Reviewer 3 Report

The modification is reasonable.